# Wetland Resource Use and Conservation Attitudes of Rural vs. Urban Dwellers: A Comparative Analysis in Thohoyandou, Limpopo Province, South Africa

**Ndidzulafhi Innocent Sinthumule** [1,*] and **Khathutshelo Hildah Netshisaulu** [2]

[1] Department of Geography, Environmental Management and Energy Studies, University of Johannesburg, Auckland Park, Johannesburg 2006, South Africa

[2] Department of Geography and Environmental Sciences, University of Venda, Private Bag X 5050, Thulamela 0950, South Africa; khathu.mathivha@univen.ac.za

[*] Correspondence: isinthumule@uj.ac.za

**Abstract:** Although wetlands provide a variety of goods and services to people and ecosystems, they are the most threatened ecosystem in the world because they are easily degraded. Thus, efforts to protect the remaining wetlands are critical if this resource is to continue providing environmental, cultural, and economic goods and services. Central to the conservation and management of wetlands is understanding the attitudes of the people bordering wetlands. This study aimed to analyse wetland resource use and conservation attitudes of urban vs. rural dwellers of Thohoyandou, Limpopo Province, South Africa. Interview-administered questionnaires and observation were the methods used for data collection. Systematic random sampling was used to obtain a sample of 282 in urban and 312 households in rural areas. Descriptive statistics (frequencies, percentages) were used to summarize the data. Chi-square ($\chi^2$) tests were applied using Microsoft Excel (Microsoft Corporation, Redmond, Washington, WA, USA) in order to determine whether responses occurred with equal probability. Differences were considered to be significant at $p \leq 0.05$. The study found that wetlands are more important in the lives and livelihoods of people in rural areas (92.9%) compared with urban areas (26.6%) of Thohoyandou. Human land use activities in wetlands (especially cultivation and infrastructural development) have degraded and destroyed wetlands—particularly those located in urban areas. The attitudes of respondents were generally positive both in urban and rural areas regarding the need for conservation and rehabilitation/restoration of wetlands. The study concluded that positive attitudes alone are insufficient to save and protect the wetlands. The responsible authorities should implement existing legislation to complement the positive attitudes of people and, importantly, they should work with communities towards the conservation of wetlands.

**Keywords:** conservation attitudes; legislation; wetland conservation; local communities; restoration; cultivation

## 1. Introduction

'Wetlands are areas of marsh, fen, peatland or water, whether natural or artificial, permanent or temporary, with water that is static or flowing, fresh, brackish or salt, including areas of marine water the depth of which at low tide does not exceed six metres' [1] (p. 9). They occupy 6% (5.7 million km$^2$) of the earth's land surface [2]. Wetlands provide many direct and indirect ecosystem services around the globe. Some of the direct benefits that they provide include storing water for domestic and irrigation purposes, serving as sources of fish, providing habitat for wetland-dependent species [3,4], and providing ecotourism opportunities and recreation purposes [5–7]. Some of the indirect benefits of wetlands include filtering pollutants [8], containing runoff, and reducing flood risk [9]. In addition, wetlands play a significant role in storing carbon and regulating the climate [10,11]. As a result, Mitsch and Gosselink [12] describe wetlands as 'the kidneys of the landscape'

because of the functions they perform, and as 'biological supermarkets' because of the socioeconomic benefits they provide to local communities.

As documented by many scholars, wetlands serve as a source of livelihood for many communities, particularly in developing nations [7,13–15], and have been important to people since early human civilisation [12]. Even though wetlands provide valuable socioeconomic and ecosystem services, it is estimated that about 30–90% of the world's wetlands have already been destroyed or have been strongly modified by anthropogenic activities [2,16]. A review of 189 reports dealing with change in coverage by wetlands found that 64–71% of wetlands had been lost by the 21st century. The review found that losses of wetlands have been more extensive and rapid for inland areas compared with coastal areas [17]. The most common anthropogenic activities affecting wetlands include agriculture, urbanisation, draining wetlands for development, invasive species, damming and water extraction, fisheries, pollution, overgrazing, industrial development, and mining activities [18–21]. Wetland destruction and degradation not only affects ecosystem functioning and health, but also those who directly depend on wetlands—particularly local communities [1,13].

As Beuel et al. [19] have noted, vegetation, soil and water quality in wetlands respond most sensitively to any land use activity. Thus, any slight disturbance of vegetation, soil, and water will have implications for wetlands. This is because wetlands are sensitive ecosystems [22]. A large body of research has evaluated anthropogenic impacts on wetlands, some of which has compared rural and urban contexts to assess changes in species diversity [23,24] and water/soil pollution [25–27]. Knowledge of changes in species and water quality is crucial if the remaining wetlands are to be properly protected and managed. Over the last two decades, scholarship focusing on both rural and urban wetlands has also evolved, particularly in developing nations. For instance, Falfushinska et al. [28] assessed the effect of carbamate fungicide on responses of biochemical markers in frogs (*Rana ridibunda*) both in rural and urban wetlands in Ukraine; more recently, Umulisa et al. [29] evaluated dichlorodiphenyltrichloroethane (DDT) residues and other persistence organic pollutants in urban vs. rural wetlands in Rwanda. In the same manner, Nortey et al. [30] carried out a comparative analysis of mangrove biomass and fish assemblages in urban and rural mangrove wetlands in Ghana. A further example is the geochemical assessment of heavy metal contamination by Ita and Anwana [31] conducted in rural and urban wetlands in Nigeria.

The literature also suggests that psychometric data, such as perceptions, beliefs, attitudes, or values, have proven to be important in the conservation and management of wetlands [22,32,33]. In the relatively limited body of research comparing rural and urban wetlands in terms of perceptions and attitudes, Hassan et al. [34] examined wetland conservation preferences among urban and rural dwellers in Malaysia, while Lannas and Turpie [35] valued the provisioning services of wetlands, contrasting a rural wetland in Lesotho with a peri-urban wetland in South Africa. To our knowledge, the article by Lannas and Turpie is the only study that has compared rural and urban wetlands in Southern Africa. The current study contributes to this developing body of knowledge on rural and urban wetlands. This study aimed to assess the resource dependence and conservation attitudes of local communities in rural vs. urban wetlands. It thus sought to identify preferences in wetland resource use and protection in rural vs. urban settings. The specific research questions were the following: What are the differences in the relative frequency of statements by the interviewees concerning the importance of wetland functions/wetland ecosystem services in rural vs. urban areas of the study area? What are the attitudes to wetland conservation of local communities in rural and urban areas? To answer these research questions, the study uses wetlands in rural and urban areas of Thohoyandou, Limpopo Province, South Africa as the case study.

## 2. Methods

### 2.1. Study Area

2.1.1. Location and Description of Thohoyandou

Thohoyandou (22°58′16.0″ S; 30°27′19.7″ E) falls under Thulamela Local Municipality—one of the four local municipalities of Vhembe District Municipality (Figure 1). The Thulamela Local Municipality was established in 2000 under the provisions of the Local Government Municipality Structures Act, 117 of 1998 Section 12 [36]. Thohoyandou is situated 180 km northeast of Polokwane, the capital of Limpopo Province, and 70 km east of Makhado town [37]. Thohoyandou town, established in the late 1970s, is situated on the main road between Makhado and the Kruger National Park. It was the capital of the Venda Bantustan (Bantu 'homeland') that was proclaimed under the Bantu Homelands Constitution Act 21 of 1971 as a self-governing territory [37]. The Venda homeland was thereby established by the apartheid government for the Venda people, that is, speakers of the Venda language. The idea of Bantustans was to exclude the majority of the Black population from the South African political system under the policy of racial segregation [38]. With the end of apartheid in 1994 and the start of democracy, the homelands were dissolved and reintegrated into provinces demarcated by the democratic government [37,39]. After 1994, Thohoyandou town nevertheless remained the main town of Thulamela and Vhembe Municipalities.

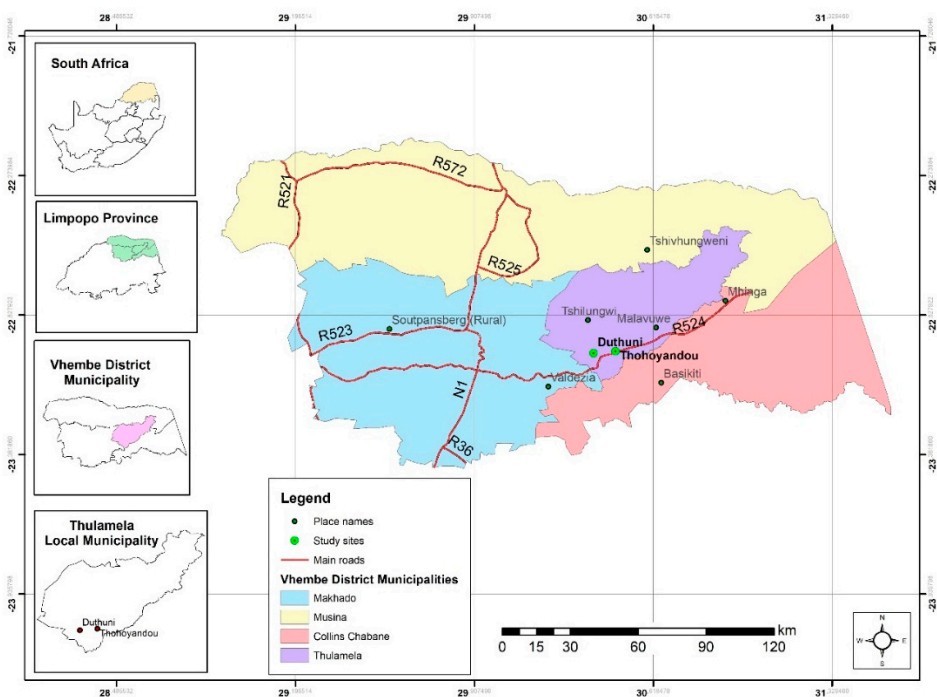

**Figure 1.** Location of Thohoyandou and Duthuni village in Thulamela municipality, both of which fall under the Vhembe district in Limpopo Province.

Thohoyandou was the site chosen for the urban sample of this study. Thohoyandou town is the fastest-growing town in Limpopo, even though there are no manufacturing industries located in the precinct. The local economy is centred on agriculture and ecotourism [40], and it is the main administrative town in Thulamela and Vhembe Municipalities. The broader area of Thohoyandou area has a population of 69,453 [40], while Blocks F and G (the focus of the urban portion of this study) have a population of 5130 distributed across 1088 households [41]. After 1994, many people moved from the rural to the urban areas of Thohoyandou in search of employment, education, and better healthcare facilities; this resulted in a high demand for land. The land in Thohoyandou town and the surrounding areas is state-owned land and is administered by the Thulamela

Local Municipality. It comprises gentle terrain, becoming hillier to the north. These hills give rise to areas that can be classified as hillslope seepage wetlands. The wetlands in the study area are interconnected and form tributaries that flow into the Luvuvhu River and supply the Nandoni Dam—the biggest dam in Thulamela, covering 1650 hectares and with a capacity of 166,200,000 m$^3$ [42] (Figure 2). Although there are few areas of wetlands that are permanently waterlogged, they mostly only become saturated after heavy rainfall. As a result, some areas of wetlands, particularly those that are not permanently flooded, are used for agriculture, and some portions have been allocated for residential purposes, thus reducing the extent of wetlands in the area [43]. Thus, over the years, human dependence and development on wetlands in Thohoyandou have contributed to wetlands degradation.

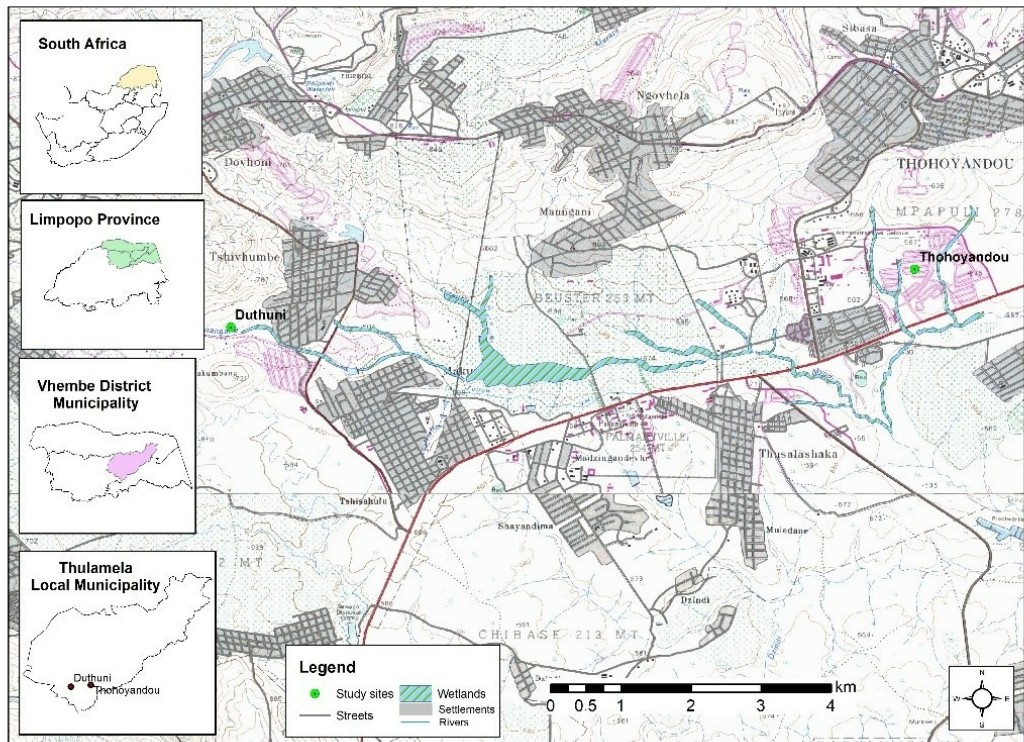

**Figure 2.** Location of the wetlands in urban vs. rural areas of Thohoyandou in the Vhembe district, Limpopo Province.

### 2.1.2. Location and Description of Duthuni

Duthuni was the location chosen for the rural sample of this study. Duthuni is a rural village (22°58′19.36″ S; 30°22′52.98″ E) located within Thulamela Local Municipality, falling under the Vhembe district in Limpopo Province of South Africa (Figure 1). Duthuni village is 19 km away from Thohoyandou town. The well-known Tshivhase Tea Estate and Vhutanda sacred natural sites are all located within Duthuni village. The latter village forms part of the Soutpansberg Mountain Range in the Vhembe Biosphere Reserve [38]. The village is on communal land that falls under the Tshivhase Tribal Council. This means the land is nominally owned by the state and the local chief is responsible for the administration and allocation of parcels of land for specific uses by individuals or organisations [44,45]. Duthuni village covers an area of 6.7 km$^2$ with a population of 6345 people distributed across 1655 households [41]. The primary land use activities in Duthuni village include settlement, subsistence agriculture, and livestock farming. Duthuni village has high levels of unemployment with a dependency on pension and social grants [22]. The groundwater discharge that gives rise to the hillslope seepage wetlands in Duthuni originates from the hills of Duthuni and Phiphidi. The flows through the wetlands are supplemented by surface water from rains. The interconnected wetlands in Duthuni village form a tributary that flows into the Madanzhe stream that then joins the Luvuvhu River which in turn

supplies the Nandoni Dam (Figure 2). Most of the wetland area remains permanently inundated because of a small dam that recharges a stream. Local communities in Duthuni have relied on wetlands for decades and they continue to rely on wetlands in the 21st century to meet their livelihoods. Human dependence is the main factor affecting the functioning of wetlands in Duthuni.

*2.2. Data Collection*

Data were collected through interview-administered questionnaires and observations. The information gathered included household demography, respondents' knowledge on wetland resource use and benefits, and their attitudes towards wetland conservation and management. In line with Jacoby and Matell [46], a three-point Likert scale (with the Likert points being 'Positive', 'Neutral', and 'Negative') was used in this study to measure the attitudes of informants. Questionnaires were translated by the bilingual authors into Tshivenda, the local language; they were then administered by three native speakers (with honours degrees) who were trained as enumerators. The data collection was thus undertaken with the help of trained research assistants under the guidance of the authors. Questionnaires were pretested on 3 field assistants and 15 selected participants who were not part of the study [47,48]. The pretesting of the questionnaire was crucial to see whether the questions were unambiguous; such testing revealed that the questions were clear and ready to be administered. All respondents were informed about the aim and scope of the research, and they were asked for their consent before they were interviewed. Interviewees were informed that their participation was voluntary, that they would not be remunerated, and that their contribution would remain anonymous.

The questionnaire interviews were administered to household heads—that is, an individual male or female who assumed the responsibility for the household [49]. Interviews were conducted in a face-to-face format during daylight hours and on weekdays, and all COVID-19 modus operandi of wearing gloves and facemasks, and keeping a distance of 1.5 m from informants, were observed throughout the research process. If the household head or anyone above the age of 18 was not present, then a second attempt was made on weekends (Saturday and Sunday). The total response time was approximately 20–30 min. In both Thohoyandou and Duthuni village, systematic random sampling was used to select households. Systematic random sampling is the sampling method of selecting samples based on specific intervals. This sampling method was used to reduce the potential for human bias and to allow a more authentic representation of the overall population [50]. To avoid interviewing more than one person from the same household, each household was taken as a sample. The number of households surveyed was determined by the respective average household size in the study area. In Thohoyandou, questionnaires were administered in every third household, whereas in Duthuni village, questionnaires were administered in every fifth household. A total of 312 households were surveyed in Duthuni village from the target households of 1655 (95% confidence level; 5% margin of error) from July to October 2020. Similarly, a total of 282 households was surveyed in Thohoyandou from a target of 1088 (95% confidence level; 5% margin of error) households from August to September 2021. In line with Fellows and Liu [47], observation was also used as a data collection technique to gain a better picture of what was taking place in situ in the study area, rather than only relying on second-hand accounts. Observation was carried out at the respective wetlands in urban and rural areas of Thohoyandou. This technique was thus used by authors to check the primary land use activities happening in the wetlands and how those activities have affected the wetlands in the study area; it was further used to observe any new development taking place in these wetlands. Observation was carried out from July to October 2020 and from August to September 2021. All the data that were observed were recorded in a notebook by the authors.

### 2.3. Data Analysis

The collected survey data were tabulated in Microsoft Office Excel 2016 (Microsoft Corporation, Redmond, Washington, WA, USA), and all analyses were conducted using Statistical Package for Social Sciences (SPSS) version 25 for Windows (IBM SPSS Inc, Chicago, IL, USA). Descriptive statistics (frequencies and percentages) were used to summarise the data. Thus, household data, data on knowledge and wetland resources use, and community attitudes towards wetlands conservation were measured using questions with three possible responses, i.e., 'Positive', 'Neutral', and 'Negative'. Regarding conservation attitudes towards wetlands, 13 related response questions were used, and open-ended questions were also provided next to each question, giving respondents the opportunity to indicate why they had made the choice they did. Individual responses to all the questions in both Thohoyandou and Duthuni village were then converted to percentages using SPSS. Chi-square ($\chi^2$) goodness-of-fit tests were applied using Microsoft Excel in order to determine if responses occurred with equal probability. Differences were considered to be significant at $p \leq 0.05$.

## 3. Results

### 3.1. Household Characteristics

Interview questionnaires were administered to 282 households in Thohoyandou and the questionnaire sample consisted of 124 males (44%) and 158 females (56%) ($\chi^2 = 4.09$, df = 1, $p < 0.043$). In Duthuni village, the questionnaires were administered to 312 households; the sample consisted of 205 women (65.7%) and 107 men (34.3%) ($\chi^2 = 30.78$, df = 1, $p < 0.05$) (Table 1).

**Table 1.** Socioeconomic profile of the respondents in Thohoyandou town (*n* = 282) and Duthuni village (*n* = 312) in Limpopo Province of South Africa.

| Categories | Class | Thohoyandou | Duthuni |
|---|---|---|---|
| | | % | % |
| Age | 18–30 years | 17.7 | 27.9 |
| | 31–40 years | 25.5 | 21.5 |
| | 41–50 years | 21.3 | 20.5 |
| | 51–60 years | 18.1 | 14.4 |
| | >61 years | 17.4 | 15.7 |
| Gender | Male | 44.0 | 34.3 |
| | Female | 56.0 | 65.7 |
| Education | None | 3.5 | 7.7 |
| | Primary | 6.7 | 13.1 |
| | Secondary | 40.4 | 53.2 |
| | Tertiary | 49.4 | 26 |
| Occupation | Unemployed | 31.6 | 66.0 |
| | Employed | 23.0 | 14.1 |
| | Self-employed | 31.6 | 10.3 |
| | Pensioner | 13.8 | 9.6 |
| Total monthly income | No income | 11.7 | 9.3 |
| | <ZAR 500 | 9.2 | 7.7 |
| | >ZAR 501–R1000 | 11.3 | 30.1 |
| | ZAR 1001–2000 | 18.8 | 20.2 |
| | >ZAR 2000 | 49.0 | 31.4 |

Because the study was conducted mostly during the day on weekdays, men were generally at work and women were the respondents for most households both in Thohoyandou and Duthuni village. Of the respondents who participated in the survey in Thohoyandou, the majority (47.2%; *n* = 133) were married, 40.1% (*n* = 113) were single, 11.7 (*n* = 33) were widows, and the remaining 1.1% (*n* = 3) did not specify (*p* < 0.05); similarly, in Duthuni

village, the majority (51.6%; *n* = 161) were single, 39.7% (*n* = 124) were married, 7.7% (*n* = 24) were widowed, and the remaining 1% (*n* = 3) did not specify ($p < 0.05$). In Thohoyandou, a total of 82.7% (*n* = 233) of the respondents had stayed in the study area for more than 11 years, and the remaining 17.4% (*n* = 49) had stayed for less than 10 years ($p < 0.05$); whereas in Duthuni village, 81.7% (*n* = 255) had stayed in the study area for more than 11 years and the remaining 18.3% (*n* = 57) had stayed for less than 10 years ($p < 0.05$). The number of people per household ranged from 5 to 10 in both Thohoyandou and Duthuni village. Other socioeconomic characteristics of respondents in Thohoyandou and Duthuni village, such as age, education, employment, and total monthly income, are represented in Table 1. It is of interest to note that 66% of people were found to be unemployed in Duthuni village ($p < 0.05$), compared with 31.6% in Thohoyandou ($p < 0.05$). Thus, the level of unemployment is different in urban and rural environments, with the rural area having a high level of people who are unemployed. In addition, about 49.4% in Thohoyandou had tertiary education ($p < 0.05$) as compared with only 26% in Duthuni village ($p < 0.05$). Furthermore, the majority of people in Thohoyandou (49%) had an income of more than ZAR 2000, as compared with Duthuni village ($p < 0.05$), which had only 31.4% at this income level ($p < 0.05$).

*3.2. Knowledge and Wetland Resource Use*

When asked if they knew about wetlands in the study area, all respondents both in Thohoyandou and Duthuni responded that they had knowledge of wetlands and where they are situated. Importantly, the majority of interviewed respondents (88.7%; *n* = 250) knew that wetlands in Thohoyandou are not permanently waterlogged but only become inundated during heavy rainfall, whereas the remaining 11.3% (*n* = 32) were not sure ($p < 0.05$). Similarly, the majority of the respondents (80.4%; *n* = 251) in Duthuni knew that the wetlands are not permanently waterlogged, whereas the remaining 19.6% (*n* = 61) were not sure ($p < 0.05$). Information on wetland resource use and benefits differed for respondents in urban vs. rural areas of Thohoyandou (Table 2).

**Table 2.** Benefits that local people obtain from wetlands in Thohoyandou town (*n* = 282) and Duthuni village (*n* = 312) falling under the Vhembe region in Limpopo Province of South Africa.

| Benefits of Wetlands to Communities | Thohoyandou | *p*-Value | Duthuni | *p*-Value |
|---|---|---|---|---|
| | % | | % | |
| Water for drinking | 0.00 | | 53.8 | |
| Water for washing clothes, car wash, or irrigation | 6.0 | | 30.1 | |
| Harvest bulrushes, sedges, and reeds for handcraft and roofing | 0.00 | | 1.9 | |
| Crop production | 5.3 | | 2.2 | |
| Grazing land for domestic stock | 0.00 | | 3.8 | |
| Important for fishing | 0.00 | | 1.0 | |
| Help to control floods | 15.3 | | 0.0 | |
| No benefit | 73.4 | <0.05 | 7.1 | <0.05 |

The majority of the respondents in Thohoyandou (73.4%; *n* = 207) observed that they did not benefit from wetlands, compared with only 7.1% (*n* = 22) in Duthuni village. The study also showed that the majority of respondents in Duthuni village (53.8%; *n* = 168) rely on water from wetlands for drinking purposes, whereas people in Thohoyandou do not rely on water from wetlands for this purpose. The findings show that, although people have municipal taps in Duthuni, they rely on water from wetlands because the municipal taps remain dry throughout the year, whereas people in Thohoyandou have taps with running water. In addition, 30.1% (*n* = 94) in Duthuni village rely on water from the wetlands for irrigation or washing clothes and cars, as compared with only 6.0% (*n* = 17) in Thohoyandou. Only 1.9% (*n* = 6) in Duthuni village harvest bulrushes, sedges, and reeds for handcraft and roofing, whereas in Thohoyandou, respondents indicated that they are not involved in harvesting wetland resources. The study also found that 5.3%

(*n* = 15) of respondents use wetlands for cultivation in Thohoyandou as compared with only 2.2% (*n* = 7) in Duthuni village. Cultivation in wetlands was found to be the major factor contributing to wetland destruction and degradation, particularly in Thohoyandou where wetlands are not permanently waterlogged (Figure 3). The main crops that were reported to be planted included maize, vegetables, and sugar cane. While 3.8% (*n* = 12) and 1.0% (*n* = 3) of respondents in Duthuni village use the wetlands for grazing and fishing purposes, respectively, people are not involved in these activities in Thohoyandou. Rather, 15.3% (*n* = 43) of respondents in Thohoyandou view wetlands as significant for controlling floods, a benefit that is not viewed as significant by all respondents in Duthuni village.

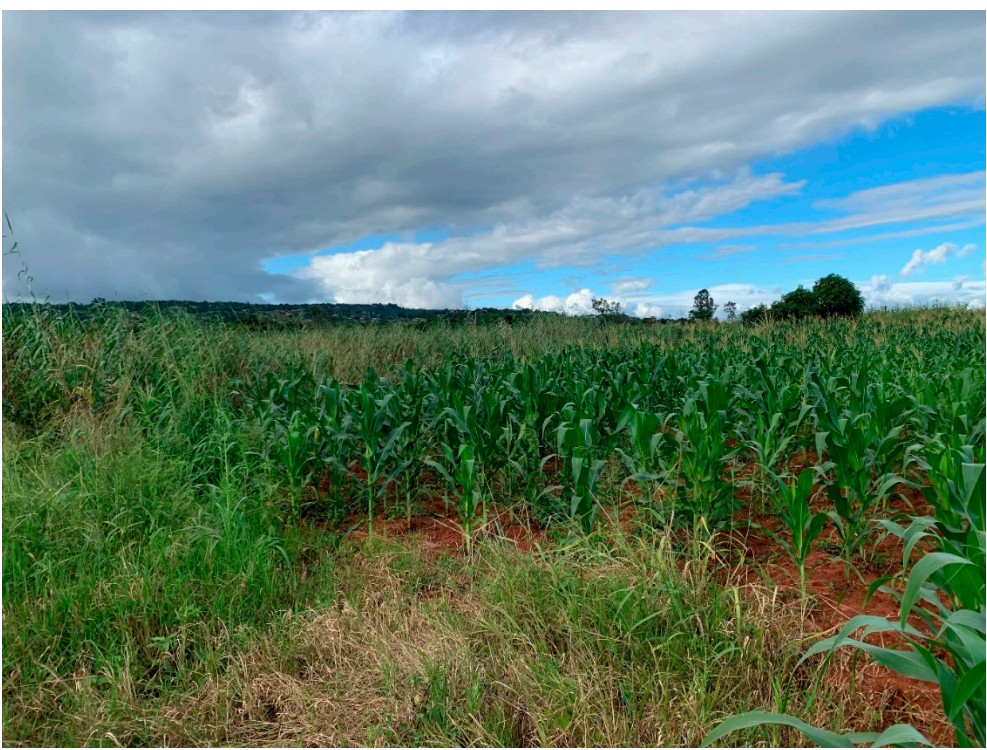

**Figure 3.** Cultivation in wetlands in the study area.

### 3.3. Attitudes to Conservation

The study found that 72.3% of respondents in Thohoyandou ($\chi^2$ = 199.97, df = 2, $p < 0.05$) were generally positive concerning the need to protect wetlands wherever they are found, compared with 98.1% in Duthuni village ($\chi^2$ = 588.59, df = 2, $p < 0.05$). The level of satisfaction with living in an area with wetlands was 55.7% in Thohoyandou ($\chi^2$ = 65.68, df = 2, $p < 0.05$) and even higher 89.4% in Duthuni village ($\chi^2$ = 442.52, df = 2, $p < 0.05$). Regarding the perception that wetlands exist for the betterment of the community, there was a 55.7% positive response for Thohoyandou ($p < 0.05$) and 93.3% for Duthuni village ($p < 0.05$) (Table 3). When local communities were asked if they were willing to donate money that can be used to protect wetlands, 63.1% were positive in Thohoyandou ($\chi^2$ = 114.32, df = 2, $p < 0.05$), as compared with 67.6% in Duthuni village ($\chi^2$ = 183.02, df = 2, $p < 0.05$). A total of 63.1% respondents in Thohoyandou ($\chi^2$ = 129.28, df = 2, $p < 0.05$) and even higher (83.0%) in Duthuni (83.0%; $\chi^2$ = 346.56, df = 2, $p < 0.05$) were also willing to go the extra mile of voting for a councillor who promised to protect wetlands. Other dominant positive responses related to the need to impose penalties on people who cause wetland destruction with 67% in Thohoyandou ($\chi^2$ = 150.53, df = 2, $p < 0.05$), compared with 92.9% in Duthuni village ($\chi^2$ = 499, df = 2, $p < 0.05$).

**Table 3.** Attitudes of community respondents towards wetland conservation in Thohoyandou (*n* = 282) town and Duthuni village (*n* = 312) in Limpopo Province of South Africa.

| Conservation Attitude Questions | Thohoyandou (%) | | | Duthuni (%) | | |
|---|---|---|---|---|---|---|
| | + | 0 | − | + | 0 | − |
| 1. I am satisfied that my village/town is located near an area that has wetlands. | 55.7 | 18.4 | 25.9 | 89.4 | 3.2 | 7.4 |
| 2. Do you approve that the wetlands exist for the betterment of people in this area? | 55.7 | 23.4 | 20.9 | 93.3 | 3.8 | 2.9 |
| 3. Do you approve that wetlands should be protected wherever they are found? | 72.3 | 20.2 | 7.4 | 98.1 | 1.6 | 0.3 |
| 4. Will you vote for a councillor who promised to protect wetlands? | 63.1 | 28.4 | 8.5 | 83 | 8 | 9 |
| 5. I am willing to donate money that can be used to protect wetlands. | 63.1 | 15.2 | 21.6 | 67.6 | 6.4 | 26 |
| 6. Do you think that the actions of local people resulted in the conservation of wetlands? | 16.3 | 26.6 | 57.1 | 59 | 6.4 | 34.6 |
| 7. Penalties should be enforced on individuals who destroy wetlands in this area. | 67.0 | 22.7 | 10.3 | 92.9 | 3.8 | 3.2 |
| 8. Land use activities that destroy or degrade wetlands should be controlled. | 67.0 | 23.0 | 9.9 | 85.9 | 9.9 | 4.2 |
| 9. Parts of wetlands that have been destroyed or degraded by anthropogenic activities should be rehabilitated. | 70.6 | 19.9 | 9.5 | 96.2 | 3.2 | 0.6 |
| 10. Rehabilitation of wetlands is a waste of money when local people are poor and short of land. | 23.8 | 28.0 | 48.2 | 17 | 7.1 | 76 |
| 11. There are laws in South Africa protect wetlands destruction and degradation? | 23.4 | 29.1 | 47.5 | 42.6 | 22.8 | 34.6 |
| 12. Government has played no role in raising awareness towards wetlands conservation and management. | 31.2 | 45.4 | 23.4 | 48.7 | 26.3 | 25 |
| 13. Overall, do you like wetlands? | 73.0 | 15.2 | 11.7 | 93.6 | 2.2 | 4.2 |

The study also found that 67.0% in Thohoyandou ($\chi^2$ = 151.29, df = 2, *p* < 0.05), compared with 85.9% in Duthuni village ($\chi^2$ = 389.48, df = 2, *p* < 0.05), were of the view that agriculture or any other land use activity that is destructive around wetlands should be controlled. Thus, despite the particular shortage of land for agriculture in urban and rural areas of Thohoyandou, the majority of respondents did not support the conversion of wetlands into agricultural lands. Instead of converting wetlands into agricultural land, 70.6% of respondents in Thohoyandou ($\chi^2$ = 180.40, df = 2, *p* < 0.05), compared with 96.2% in Duthuni village ($\chi^2$ = 554.38, df = 2, *p* < 0.05), were of the view that degraded wetlands should be rehabilitated. When respondents were asked whether their actions had resulted in the conservation of wetlands, 57.1% of respondents in Thohoyandou said 'no' ($\chi^2$ = 76.11, df = 2, *p* < 0.05), as compared with 34.6% in Duthuni village ($\chi^2$ = 74.15, df = 2, *p* < 0.05). The response by Thohoyandou respondents is understandable because those wetlands have been seriously degraded by anthropogenic activities (notably cultivation and infrastructural development).

Although the wetlands of Duthuni village have also been degraded by cultivation, they are in a better condition compared with those in Thohoyandou. As a result, credit should be given to local communities for the role they have played in protecting these resources. When asked whether rehabilitation of wetlands is a waste of money when local people are poor and short of land, almost half of the respondents (48.2%) said 'no' in Thohoyandou ($\chi^2$ = 28.91, df = 2, *p* < 0.05), and a higher proportion of 76% was found in Duthuni village ($\chi^2$ = 259.75, df = 2, *p* < 0.05). The study also found that 47.5% of respondents in Thohoyandou were not aware that laws in South Africa protect wetlands ($\chi^2$ = 26.89, df = 2, *p* < 0.05), as compared with 34.6% in Duthuni village ($\chi^2$ = 18.71, df = 2, *p* < 0.05). When asked whether the government has a role in raising awareness towards wetlands conservation, 45.4% were not sure in Thohoyandou ($\chi^2$ = 21.02, df = 2, *p* < 0.05),

compared with 26.3% in Duthuni village ($\chi^2$ = 33.31, df = 2, $p$ < 0.05). Overall, 73% of respondents reported appreciating wetlands in Thohoyandou, and had positive attitudes towards wetlands ($\chi^2$ = 200.70, df = 2, $p$ < 0.05), as compared with 93.6% in Duthuni village ($\chi^2$ = 509.94, df = 2, $p$ < 0.05).

## 4. Discussion

The study found that the type of wetlands that are found in the study area are hillslope seepage wetlands that originate from hills. These wetlands are located on the mid- and footslopes of hillsides and originate from springs where groundwater emerges at the ground surface [51]. As Kotze [52] has noted, seepage wetlands are usually connected to valley bottom wetlands or rivers as in the study area. The results of this study show that the majority of the respondents in Duthuni village rely on wetlands daily. Thus, without wetlands, local communities cannot meet their daily needs. The important socioeconomic benefits that rural people of Duthuni derive from wetlands include water for domestic use and irrigation, crop production, fishing, cattle grazing, and harvesting of plant resources for roofing and handcraft production. Wetlands are also reported to provide an extensive range of direct benefits to rural communities in various parts of the world, including in Iran [7], the Ghodaghodi Lake area in Nepal [53], Kabartal in India [54], the Central Rift Valley of Ethiopia [55], and Trinidad in South America [56]. Thus, wetland benefits serve as a means of supporting life and livelihoods for rural communities [57], particularly in developing nations [58]. For instance, wetlands are home to flora species used to make different handicrafts products that serve as a source of livelihood for many communities [13]. Local communities also make an income from the sale of some of the resources (e.g., fish, edible crops) harvested from wetlands [59]. In addition, wetlands have variety of plants and animal species that have medicinal and ecotourism value [60]. Although rural communities in Duthuni village rely on the local wetlands for their survival, the cultivation of maize, peppers, and other vegetables was found to be the major factor contributing to wetland degradation and destruction. These results are consistent with Dixon and Wood [61], who also found wetland cultivation to be the major factor contributing to wetland destruction in Eastern Africa. Similarly, Song et al. [62] also found that agriculture accounted for 91% of wetland losses in the Sanjiang Plain in China. This study also found that urban wetlands in Thohoyandou are also critical in the lives of local communities. Although the majority of people do not benefit directly from wetlands, 15.3% of respondents indicated that wetlands are important to them for controlling floods. This is in line with Pattison-Williams et al. [63], who argued that the retention of existing wetlands is an economically viable means to limiting the financial, social, and environmental damages of flooding. According to Jisha and Puthur [64], wetlands act like natural sponges that collect and store water from heavy rains and rapid snowmelts; they release water slowly, thereby reducing the damage from seasonal (and often catastrophic) floods. The study also found that few respondents rely on potable water from urban wetlands and instead use this for purposes such as car washing or irrigation; importantly, however, they also use wetlands for cultivation. The wetlands in Thohoyandou are particularly vulnerable to cultivation because they are not permanently waterlogged. This characteristic has allowed local communities to cultivate maize, vegetables, and sugarcane in the local wetlands (Figure 4). The use of urban wetlands for cultivation is not exclusive to the study area; Mandishona and Knight [32] also found that wetlands in urban areas of Harare (Zimbabwe) are commonly used for cultivation purposes. Similarly, a peri-urban wetland in Mfuleni, Cape Town, was also found to be used for cultivation by local communities [35]. As in the present study area, urban wetland cultivation was also found to be the major factor contributing to wetland destruction [32]. Although agriculture in wetlands is a source of income for both rural and urban dwellers, it is also a source of wetland degradation. Thus, human activities are the major factor affecting wetlands in the study area. This has negative implication for the ecological functioning of wetlands. Given that wetlands in Thohoyandou are not permanently inundated, and despite the government restrictions in this regard, the local

municipality, over the past 10–15 years, has allocated some business and residential plots within wetlands for development (Figure 5). This has contributed to wetland destruction and degradation.

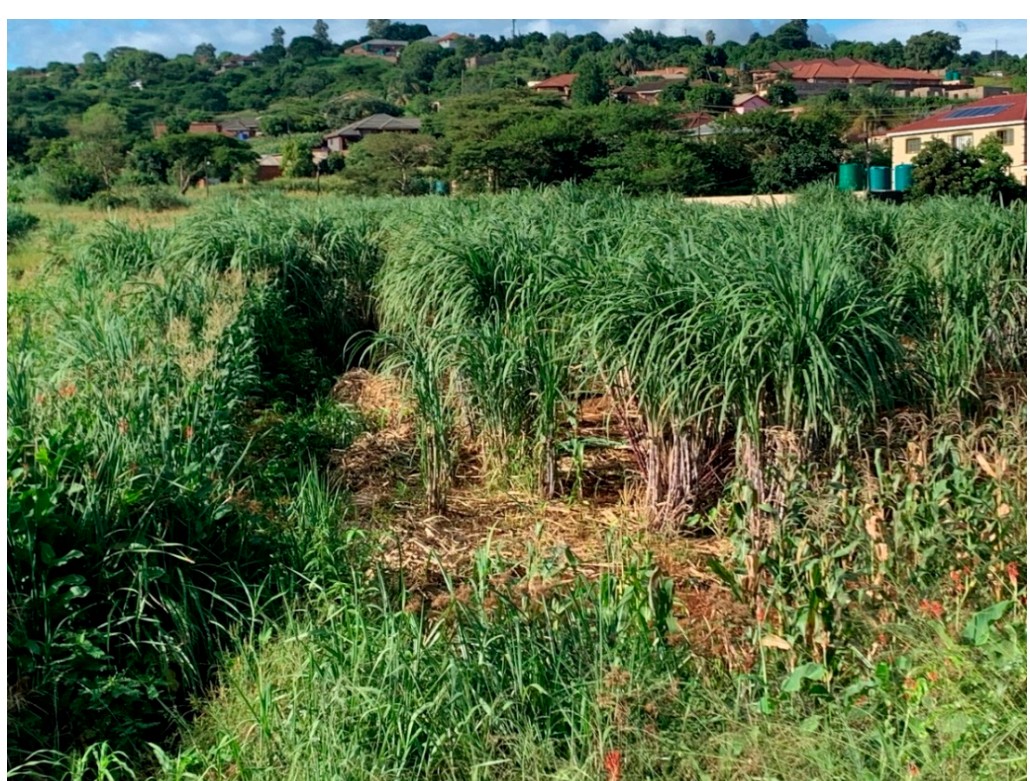

**Figure 4.** Sugarcane cultivated on wetlands in Thohoyandou.

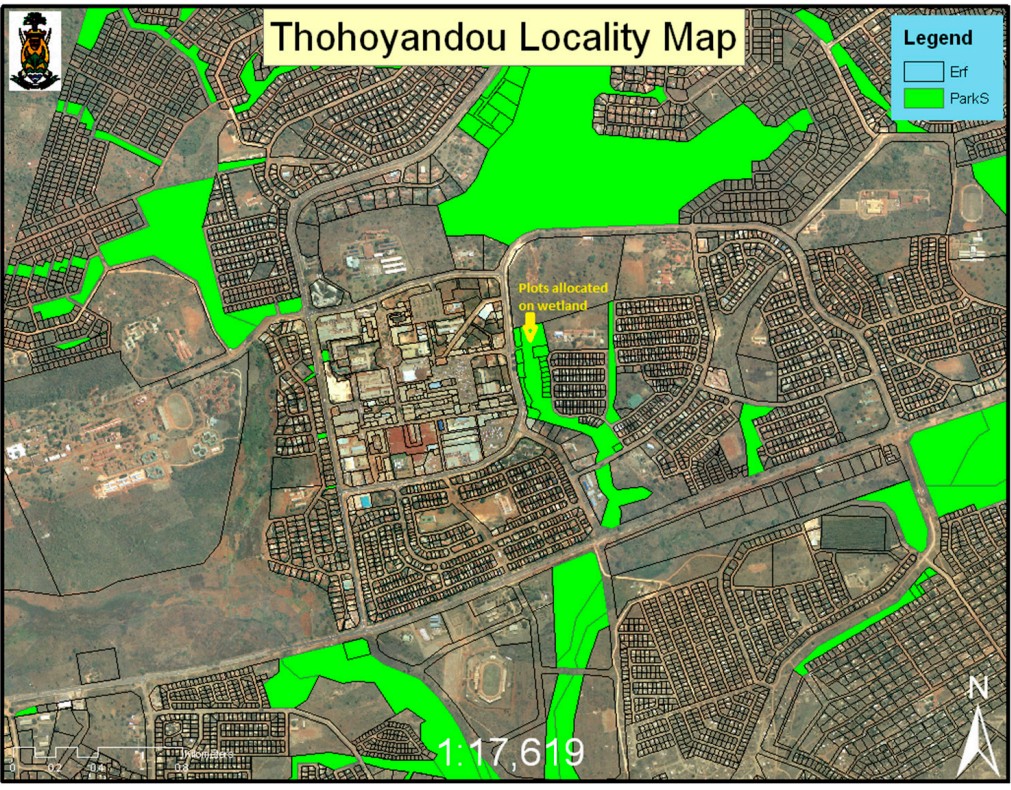

**Figure 5.** Allocation of residential and business stands on wetlands in Thohoyandou.

Infrastructural development has already destroyed large portions of wetlands around the world and many areas of wetlands are degraded as in the case of Nakivubo urban wetland in Uganda [65] and Lagos wetlands in Nigeria [66]. A similar pattern has occurred in South Africa despite wetlands being protected by a number of regulations, including the Conservation of Agricultural Resources Act of 1984, the National Environmental Management Act 107 of 1998, the National Water Act 36 of 1998, and the Environmental Provisions of the Mineral and Petroleum Resources Development Act 28 of 2002. Although South Africa has progressive wetland laws and regulations, only 23.4% in Thohoyandou and 42.6% in Duthuni village are aware that there are laws that protect wetlands. As is the case in Zimbabwe [67], the limited awareness can be ascribed to insufficient education and enforcement by responsible government institutions.

The destruction of wetlands is a cause for concern in both urban and rural areas. As a result, efforts are made to protect and conserve the remaining wetlands in various parts of the world. In the case of the study area, the questionnaire was designed to assess the attitudes of local communities in urban and rural dwellers of Thohoyandou. A wealth of past research has analysed differences in attitudes towards the environment and environmental protection between urban and rural populations [68–70]. There is no conclusive evidence that nature conservation or protection of the environment is favoured more by people in the urban or rural areas. In the case of the study area, responses were positive in urban and rural dwellers concerning the protection of wetlands, satisfaction with living in an area with wetlands, and the perception that wetlands exist for the betterment of the community. Unlike in the Setiu wetlands in the state of Terengganu in Malaysia, where rural people were unwilling to pay for wetland conservation, while urban people favoured wetland conservation and exhibited a preference for wetland attributes [34], in the study area, the majority of urban and rural respondents were willing to donate money that could be used to protect wetlands. Furthermore, urban and rural respondents in the study area showed they would be willing to vote for a local politician who promised to protect the wetlands. This mirrors similar findings in Oxfordshire and Cambridgeshire in the United Kingdom [71]. Other positive responses by the majority of urban and rural dwellers surveyed in the study relate to the protection of wetlands from agriculture or any other land use that may destroy wetlands. Thus, despite people having insufficient land for agriculture, they voiced the fact that they did not support the draining of wetlands, or the encroachment of land uses that may destroy or degrade wetlands.

These results are consistent with Bikangaga et al. [72] and Mandishona and Knight [32] elsewhere in Uganda and Zimbabwe. However, they conflict with other studies—for instance, those of Ambastha et al. [52], who found that most respondents demanded draining of the water to make more land available for agriculture in Kabartal Wetland in the Indo-Gangetic plains of India. In the Jimma Highlands in Southwestern Ethiopia, the majority of respondents (66%) were not interested in conserving wetlands because of the small extent of small landholdings and the need to meet their needs [73]. Similarly, Pyrovetsi and Daoutopoulos [74] found that wetland farmers had a relatively negative attitude towards the conservation of wetlands in Macedonia in Greece. Thus, people who are highly dependent on wetland resources for their livelihoods can often be reluctant to protect wetlands. However, in the case of the current study, instead of degrading or destroying wetlands, the majority of rural and urban dwellers in Thohoyandou were of the view that wetlands should be rehabilitated or restored. This is consistent with Zhang et al. [75], who found that more than half of farmer respondents (56.13%) in China were in favour of wetland restoration—regardless of their geographical location or economic interests. As Arjunan et al. [76] have noted, implementing wetland restoration—particularly in developing nations—represents a challenge because a large number of farmers depend on wetlands for their livelihoods. As a result, consultation of affected parties is critical in restoration planning [77] and, importantly, compensation of affected people should be carried out to make up for the loss of land, particularly agricultural land [76].

## 5. Conclusions

This study has demonstrated that wetlands are critical for communities in both urban and rural areas of Thohoyandou. Although wetlands are important to local communities, portions of the wetlands in the study area have been seriously degraded/destroyed as a result of agriculture. In addition, wetlands particularly in urban areas are also affected by infrastructural development. Human activities have affected the ecological functioning of wetlands. This is a cause for concern, particularly in a country such as South Africa which is a water-scarce area, and where the majority of people depend on wetlands daily. Unlike in other studies where local communities have negative attitudes towards conservation of the environment [73,78], this study has demonstrated that local dwellers in urban and rural areas of Thohoyandou have positive attitudes towards the conservation of wetlands. This includes their willingness to see wetlands restored, their preparedness to control any land use activities that may cause wetlands destruction and degradation, and their determination to pay for wetland conservation. Thus, local communities do not see wetlands as 'wasteland' that serves as a dumping ground for waste materials, as in other areas [79]; rather, they see wetlands as a valuable resource that should be protected for present and future generations. This is a positive step towards saving the remaining wetlands and rehabilitating those that are degraded. However, the positive attitudes alone are insufficient to save or protect the wetlands in the study area. Rather, there is a need for the government to work with communities in identifying wetlands that are of significance to people. Furthermore, the government needs to rehabilitate/restore wetlands in South Africa. Such restoration should start with consultation with affected communities, and where possible, those that are affected should be compensated. Importantly, there is a need for the government to implement existing legislation in South Africa to strengthen the positive attitudes of people towards wetland conservation. Local communities and government/responsible authorities working together can help to save those wetlands that are on the brink of disappearing. This approach will allow wetlands to continue providing the socioeconomic benefits and other ecosystem services that are critical for affected populations.

**Author Contributions:** Conceptualization, N.I.S.; Data curation, N.I.S. and K.H.N.; Formal analysis, N.I.S.; Investigation, N.I.S. and K.H.N.; Methodology, N.I.S. and K.H.N.; Project administration, N.I.S. and K.H.N.; Resources, N.I.S. and K.H.N.; Software, K.H.N.; Validation, N.I.S. and K.H.N.; Visualization, N.I.S.; Writing—original draft, N.I.S.; Writing—review & editing, N.I.S. and K.H.N. All authors have read and agreed to the published version of the manuscript.

**Funding:** This work was funded by internal research grants by the University of Johannesburg (URC-grant) and the University of Venda, South Africa.

**Institutional Review Board Statement:** Not applicable.

**Informed Consent Statement:** Informed consent was obtained from all subjects involved in the study.

**Data Availability Statement:** The data presented in this study are available on request from the corresponding author.

**Acknowledgments:** We acknowledge all the respondents in Thohoyandou and Duthuni who participated in this study. A special thanks to the chief of Duthuni Village (Chief Ligege) and the Traditional Council for allowing us to do this research in their village. N.I.S. gratefully acknowledge financial support from the University of Johannesburg (URC-grant).

**Conflicts of Interest:** The authors declare no conflict of interest.

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
