# Peer review of "Wetland Resource Use and Conservation Attitudes of Rural vs. Urban Dwellers: A Comparative Analysis in Thohoyandou, Limpopo Province, South Africa"

_water, doi:10.3390/w14081290_

Round 1

Reviewer 1 Report

The manuscript 'Wetland resource use and conservation attitudes of rural versus urban dwellers: A comparative analysis in Thohoyandou, Limpopo Province, South Africa' addresses the use and perception of wetlands in two communities with different population sizes. The wetlands are classified as 'hillslope seepage wetlands', receiving water from seasonal precipitation. They are distributed along drainage channels belonging to the catchment area of Luvuvhu river. Downstream from the two communities, Luvuvhu river flows into the Nandoni reservoir lake. In the larger community, the wetlands are waterlogged only for part of the year, whereas in the smaller community, higher water levels are maintained throughout the year by regulating flow rates using a small dam.

The use and the perception of the wetlands by inhabitants of the two communities were investigated through a questionnaire survey. The use patterns differed between the more urban and the more rural community, with a higher proportion of respondents stating dependence on wetland resources in the more rural setting. Attitudes towards wetlands were predominantly positive in the more rural community, whereas benefits from wetlands were perceived by a much smaller part of the local community in the more urban environment.

The topic of the study is relevant for the journal Water, the methodology of the survey is appropriate and mostly well-documented. The main aspect that needs improvement is the statistical analysis of the data. In order to answer the research question concerning differences in use patterns and perception of wetlands between the more rural and the more urban environment, results of tests of the statistical significance of differences between the relative frequencies need to be presented.

Furthermore, the issue of differences between the wetland types in Duthuni and Thohoyandou in ecological and in socio-economic terms and the potential effects of these differences on the results obtained in the survey needs to be discussed in more detail.

The other necessary revisions are minor and can be achieved by text editing:

Line 39: ... socio-economic and ecosystem services ... -> Ecosystem services are by definition 'socio-economic' entities; it is nonsensical to separate the terms.

Line 44: ... filtering pollutants, free purification of water ... -> These two statements refer to the same function associated with wetlands; one is therefore redundant. Expressions such as 'free purification' are ambiguous and should be avoided.

Line 46: ... storing the regulation of carbon and climate -> Do you mean 'storing of carbon and regulation of climate'?

Line 57: ... [13-15,7] -> Check the numbering/order of references.

Lines 63 - 65: Check for redundancies/overlap between categories.

Line 71: The statements concerning the sensitivity of wetland ecosystems to disturbance events are too broad. Not all disturbances are negative; some are actually necessary for maintaining diversity.

Line 80: ... wetlands have also ... -> ... wetlands has also ...

Line 93: Beyond mere valuation ... -> This phrase seems out of context, because valuation of wetlands was not discussed in the previous paragraphs.

Line 108: ... relative importance of wetlands ... -> This expression is problematic, because it implies that you measured the importance of wetlands for the more rural and the more urban community relative to some general metric of what is important for the respondents. What your study design allows you to do is to quantify differences in the relative frequency of statements by the interviewees concerning the importance of wetland functions/wetland ecosystem services.

Line 110: ... indigenous and local communities ... -> This suggests that you distinguished between indigenous and local respondents; because you don't separate 'indigenous' and (non-indigenous) 'local' participants in the survey, this should be reworded.

Line 119: How do the two communities included in the survey compare to the frequency distribution of the size (number of inhabitants, area etc.) of municipalities in South Africa? Can they be considered typical in terms of these criteria.

Line 147: ... ecotourism [40] and it is ... -> ... ecotourism [40], and it is ...

Line 149: Check notation of numbers (space separating 1000)

Line 169: Legend Fig 2: Include the meaning of the grey, blue and pink shading in the map

Line 191: ... Duthuni orginate ... -> Duthuni originates ...

Line 192: ... wetlands and are ... - > ... wetlands are ...

Line 224: Please describe the selection procedure for the households in more detail. Was the random sampling conducted based on a complete list of households?

Line 245 Please describe how the observation was carried out and documented (where, when by whom, what was recorded etc.

Line 265: ..., allowing ... -> ..., giving ...

Line 300: Does the meaning/the practical relevance of 'unemployment' differ between the more rural and the more urban environment?

Line 398 ... were not aware ... / line 399: ... were aware ... -> Please use the same statistic to allow comparison.

Author Response

Reviewer 1

Dear Reviewer #1. Thank you for the useful comments you have made on my manuscript. I have addressed your comments as follows:

Comment

Response

The topic of the study is relevant for the journal Water, the methodology of the survey is appropriate and mostly well-documented. The main aspect that needs improvement is the statistical analysis of the data. In order to answer the research question concerning differences in use patterns and perception of wetlands between the more rural and the more urban environment, results of tests of the statistical significance of differences between the relative frequencies need to be presented.

·         Test of statistical significance has been done on the results section from pages 6 to 10.

Furthermore, the issue of differences between the wetland types in Duthuni and Thohoyandou in ecological and in socio-economic terms and the potential effects of these differences on the results obtained in the survey needs to be discussed in more detail.

·         This important comment has been addressed on page 10.

The other necessary revisions are minor and can be achieved by text editing:

·         Line 39: ... socio-economic and ecosystem services ... -> Ecosystem services are by definition 'socio-economic' entities; it is nonsensical to separate the terms.

·         The word socio-economic has been removed.

·         Line 44: ... filtering pollutants, free purification of water ... -> These two statements refer to the same function associated with wetlands; one is therefore redundant. Expressions such as 'free purification' are ambiguous and should be avoided.

·         The sentence has been improved and the repeating statement ‘free purification of water’ has been removed.

·         Line 46: ... storing the regulation of carbon and climate -> Do you mean 'storing of carbon and regulation of climate'?

·         Storing the regulation of carbon and climate has been replaced by 'storing of carbon and regulation of climate.

·         Line 57: ... [13-15,7] -> Check the numbering/order of references.

·         The order of references have been improved.

·         Lines 63 - 65: Check for redundancies/overlap between categories.

·         Redundancies have been removed by erasing the first part of the sentence.

·         Line 71: The statements concerning the sensitivity of wetland ecosystems to disturbance events are too broad. Not all disturbances are negative; some are actually necessary for maintaining diversity.

·         The word negative has been removed.

·         Line 80: ... wetlands have also ... -> ... wetlands has also ...

·         This has been corrected on page 2.

·         Line 93: Beyond mere valuation ... -> This phrase seems out of context, because valuation of wetlands was not discussed in the previous paragraphs.

·         Beyond mere valuation has been removed and the sentence has been rephrased.

·         Line 108: ... relative importance of wetlands ... -> This expression is problematic, because it implies that you measured the importance of wetlands for the more rural and the more urban community relative to some general metric of what is important for the respondents. What your study design allows you to do is to quantify differences in the relative frequency of statements by the interviewees concerning the importance of wetland functions/wetland ecosystem services.

·         The sentence has been rephrased as suggested by the reviewer.

·         Line 110: ... indigenous and local communities ... -> This suggests that you distinguished between indigenous and local respondents; because you don't separate 'indigenous' and (non-indigenous) 'local' participants in the survey, this should be reworded.

·         The word indigenous has been removed.

·         Line 147: ... ecotourism [40] and it is ... -> ... ecotourism [40], and it is ...

·         This sentence has been corrected.

·         Line 149: Check notation of numbers (space separating 1000).

·         This has been corrected.

·         Line 169: Legend Fig 2: Include the meaning of the grey and blue shading in the map.

·         The meaning of grey and blue shading has now been included on the map.

·         Line 191: ... Duthuni orginate ... -> Duthuni originates ...

·         This has been corrected.

·         Line 192: ... wetlands and are ... - > ... wetlands are ...

·         This has been corrected.

·         Line 224: Please describe the selection procedure for the households in more detail. Was the random sampling conducted based on a complete list of households?

·         Detail information has been added.

·         Line 245 Please describe how the observation was carried out and documented (where, when by whom, what was recorded etc.

·         Detail on how observation has been carried out is given.

·         Line 265: ..., allowing ... -> ..., giving ...

·         Allowing has been replaced by giving as suggested by reviewer.

·         Line 300: Does the meaning/the practical relevance of 'unemployment' differ between the more rural and the more urban environment?

·         This has been clarified.

·         Line 398 ... were not aware ... / line 399: ... were aware ... -> Please use the same statistic to allow comparison.

·         Same statistics have been used as suggested.

Reviewer 2 Report

However, I find that the manuscript would benefit by adding information about the history of, and the current ecologic status of the wetlands. In particular information to understand the difference in the status of the wetland in the urban and the rural area. Is the difference in the current status of the wetland in the urban versus the rural area mainly due to human development of the land in the urban area in contrast to the rural area? Or is the difference mainly because of hydrology, a natural situation. Information on why the wetlands in the rural areas have not been developed for agriculture use, is this because of law enforcement, or local norms (reflecting the informants perspectives on conservation), or other reasons.  This would have provided a needed (?), and an interesting context to the interview results that a high percentage of the informants express high appreciation of the wetlands, and the need for protection of wetlands.

Perhaps outside of the scope of this manuscript, yet interesting would be information on any local governance for protection of the wetlands.  

Author Response

Reviewer 2

Dear Reviewer #2. Thank you for the useful comments you have made on my manuscript. I have addressed your comments as follows:

Comment

Response

However, I find that the manuscript would benefit by adding information about the history of, and the current ecologic status of the wetlands. In particular information to understand the difference in the status of the wetland in the urban and the rural area. Is the difference in the current status of the wetland in the urban versus the rural area mainly due to human development of the land in the urban area in contrast to the rural area? Or is the difference mainly because of hydrology, a natural situation.

·         The information dealing with history of and current ecological status of the wetlands has been provided under the section that is dealing with the study area.

·         The information on the status of wetlands have also been given under results on page 9 and discussion on page 10.

·         Under the section dealing with discussion, it was indicated that the current status of wetlands in urban versus rural areas is mainly due human activities and not natural factors. 

Information on why the wetlands in the rural areas have not been developed for agriculture use, is this because of law enforcement, or local norms (reflecting the informants perspectives on conservation), or other reasons.  This would have provided a needed (?), and an interesting context to the interview results that a high percentage of the informants express high appreciation of the wetlands, and the need for protection of wetlands.

·         Wetlands in rural areas are also affected by agriculture which has contributed to the destruction of wetlands. This information has been provided on page 9 under results section and further discussed on page 10 under discussion section.
